# Toxicity and Behavior-Altering Effects of Three Nanomaterials on Red Imported Fire Ants and Their Effectiveness in Combination with Indoxacarb

**DOI:** 10.3390/insects15020096

**Published:** 2024-02-01

**Authors:** Zewen Ma, Jiantao Fu, Yunfei Zhang, Lanying Wang, Yanping Luo

**Affiliations:** 1School of Tropical Agriculture and Forestry, Hainan University, Haikou 570228, China; 2Institute of Nanfan & Seed Industry, Guangdong Academy of Sciences, Guangzhou 510316, China

**Keywords:** red imported fire ants, nanomaterials, toxicity, indoxacarb, compounding

## Abstract

**Simple Summary:**

The red imported fire ant is an invasive pest worldwide and a social insect that presents a significant risk to agricultural productivity, ecosystems, and human health. This work presents the selection of three nanomaterials out of a total of 16 species. We assessed the toxicity and behavioral alteration effects of these three materials individually, as well as in combination with indoxacarb, on red imported fire ants. The results showed that diatomite and multi-walled carbon nanotubes could enhance the activity of indoxacarb against red imported fire ants and inhibit their colony behavior. Diatomite and multi-walled carbon nanotubes can be added as pesticide additives to improve the control effects on red imported fire ants, reduce the use of indoxacarb, and delay the development of resistance.

**Abstract:**

The red imported fire ant (*Solenopsis invicta* Buren) is one of the 100 worst invasive alien species in the world. At present, the control of red imported fire ants is still mainly based on chemical control, and the most commonly used is indoxacarb bait. In this study, the contact and feeding toxicity of 16 kinds of nanomaterials to workers, larvae, and reproductive ants were evaluated after 24 h, 48 h, and 72 h. The results showed that the mortality of diatomite, Silica (raspberry-shaped), and multi-walled carbon nanotubes among workers reached 98.67%, 97.33%, and 68%, respectively, after contact treatment of 72 h. The mortality of both larval and reproductive ants was less than 20% after 72 h of treatment. All mortality rates in the fed treatment group were below 20% after 72 h. Subsequently, we evaluated the digging, corpse-removal, and foraging behaviors of workers after feeding with diatomite, Silica (raspberry-shaped), and multi-walled carbon nanotubes for 24 h, which yielded inhibitory effects on the behavior of red imported fire ants. The most effective was diatomite, which dramatically decreased the number of workers that dug, extended the time needed for worker ant corpse removal and foraging activities, decreased the number of workers that foraged, and decreased the weight of the food carried by the workers. In addition, we also evaluated the contact and feeding toxicity of these three nanomaterials in combination with indoxacarb on red imported fire ants. According to contact toxicity, after 12 h of contact treatment, the death rate among the red imported fire ants exposed to the three materials combined with indoxacarb reached more than 97%. After 72 h of exposure treatment, the mortality rate of larvae was more than 73% when the nanomaterial content was above 1% and 83% when the diatomite content was 0.5%, which was significantly higher than the 50% recorded in the indoxacarb control group. After 72 h of feeding treatment, the mortality of diatomite, Silica (raspberry-shaped), and multi-walled carbon nanotubes combined with indoxacarb reached 92%, 87%, and 98%, respectively. The death rates of the three kinds of composite ants reached 97%, 67%, and 87%, respectively. The three kinds of composite food had significant inhibitory effects on the behavior of workers, and the trend was largely consistent with the effect of nanomaterials alone. This study provides technical support for the application of nanomaterials in red imported fire ant control.

## 1. Introduction

The International Union for Conservation of Nature (IUCN) listed the red imported fire ant (*Solenopsis invicta* Buren) as one of the 100 most threatening invasive alien species in the world [1]. This species originated in South America and was introduced to the United States in the early 20th century. Subsequently, this species has rapidly spread to many parts of the world, invading countries like Australia, China, Japan, and South Korea [2,3]. Red imported fire ants are, therefore, a very serious threat. These ants can directly harm crops by consuming the seeds, fruits, shoots, roots, and seedlings of numerous crops, such as cucumbers, soybeans, maize, and eggplants, causing direct damage to crops [4]. In addition, the invasion of red imported fire ants has greatly reduced the abundance and diversity of native species and may even have displaced them [5]. Other native species may also be significantly impacted. Indeed, some research indicates that red imported fire ants may even change the morphology and behavior of native lizards [6,7]. Apart from causing devastation to ecosystems, red imported fire ants pose a serious threat to human safety due to their intense aggression and high toxicity [8]. People who are allergic to red imported fire ants may experience burning pain, followed by redness, swelling, itching, high fever, and, in severe cases, shock or even death [9]. Red imported fire ants also congregate around uninsulated electrical equipment, causing short circuits and interfering with regular switch operation, damaging agricultural machinery and equipment, and interfering with manual operation [10,11]. Red imported fire ants cause enormous economic harm, costing the United States more than USD 6 billion per year in damage restoration, medical care, and control measures [12,13].

The damage caused by red imported fire ants is severe and difficult to eliminate. At present, chemical pesticides remain the primary means of controlling red imported fire ants [14]. These pesticides are used primarily as stomach poisons in bait products or as contact insecticides in scatter, wet, and dust products for areal and nest treatment [15]. Because of their delayed toxicity, which allows the active component to permeate throughout the colony, killing all the red imported fire ants in the nest [16,17], insecticidal baits are a common and successful technique for controlling red imported fire ants. Indoxacarb is currently the most extensively used pesticide for controlling red imported fire ants [18]. Indoxacarb is a novel broad-spectrum oxadiazine insecticide that is insoluble in water, has low toxicity, offers rapid and strong control effects, and requires minimal application. Indoxacarb has been demonstrated to be particularly effective against several dangerous insects, including ants, cockroaches, leafhoppers, and aphids [19,20,21].

Chemical pesticides are the most common technique for controlling red imported fire ants, although the possible negative consequences of chemical insecticides are becoming well recognized. Traditional pesticide preparations have disadvantages such as a fast release of active ingredients, short duration of efficacy, poor dispersion, and easy loss. Less than 0.1% of the pesticides act on the target, and the utilization rate is extremely low [22]. Intensive use of inefficient traditional formulations of pesticides has yielded irreversible damage to the ecological environment and non-target organisms, and long-term control and use are also prone to insecticide resistance among red imported fire ants [23,24]. With the extensive use of traditional pesticide preparations, environmental pollution, resistance, and other problems may arise [25]. To reduce the use of chemical insecticides, people are exploring alternative strategies for controlling pests that are less prone to resistance and more human-friendly. The most effective approach to combat resistance is to create new insecticides with novel modes of action. However, new active chemicals are expensive to discover and develop, and it can take 10–15 years for such chemicals to enter commercial use [26]. As a result, improving pesticide formulations with high activity is a cost-effective way to delay or overcome resistance [27]. Among them, nanomaterials are an alternative strategy with good prospects.

Some nanomaterials have insecticidal properties. Organic polymer nanoparticles (NPs) including biodegradable chitosan (CS) and carboxymethyl chitosan (CMCS) show insecticidal effects on red imported fire ants. Food consumption, growth, and the development of red imported fire ants are repressed after feeding. In addition, the midgut swells noticeably, while digestive enzyme activity in the midgut diminishes [28]. NiCoNC, a hexagonal star nanoparticle grafted with carbon nanotubes, was previously obtained via high-temperature pyrolysis and reduction using the imidazole skeleton zeolite 67 (ZIF-67) as a carrier. Although NiCoNC showed no insecticidal activity against *S. invicta*, it increased the insecticidal activity against red imported fire ants by entering the gut and increasing sensitivity to the insecticide [29]. Diatomite has also been used to control red imported fire ants. Adding diatomite to fertilizer can lead to increased termite mortality [30]. Moreover, both raw diatomite and calcined diatomite have been used as natural insecticides to remove common stored-grain pests [31]. Multi-walled carbon nanotubes can enhance the toxic effects of dibutyl phthalate on zebrafish in the early life stages [32]. Silica nanoparticles can be used as insecticides to kill target pests or as carriers of insecticide molecules for sustained release [33]. Nanoparticle development also has promise for improving pesticide formulations and increasing insecticidal performance.

This study aims to evaluate the contact toxicity and feeding toxicity of nanomaterials against red imported fire ant workers and larvae by searching for nanomaterials with high activity against red imported fire ants and investigating the influence of nanomaterials on the behavior of workers and the vitality of ant colonies. In combination with indoxacarb, the activity, and effects on behavior were also assessed. The results are expected to provide pesticide additives that can improve the activity of indoxacarb and the control effects of red imported fire ants without increasing the amount of indoxacarb.

## 2. Materials and Methods

### 2.1. Nanomaterials

Multi-walled carbon nanotubes (long: 5–15 nm, 10–30 μm; short: 8–15 nm, 0.5–2 μm), graphene nanosheets, and graphite oxide were purchased from Jiangsu Xianfeng Nanomaterials Technology Co., Ltd., Nanjing, China. Nano copper oxide (40 nm, spherical; 100–200 nm, spherical), hydrotalcite, and attapulgite were purchased from Hainan Haidaosen Technology Co., Ltd., Haikou, China. Diatomite was purchased from the French Imerys Group, Villers-Sous-Saint-Leu, France. Zein, halloysite (200 nm; 500 nm), and Silica (20 nm; 80 nm; 410 nm; raspberry-shaped) were synthesized by the research team.

### 2.2. Red Imported Fire Ant for Testing

Red imported fire ants were collected in Dongge Town, Wenchang City, Hainan Province, and were raised in 25 L plastic buckets, the inner walls of which were coated with Polytetrafluoroethylene powder. The ants were regularly fed with a water–sugar solution (0.1 g/mL) using ham sausages as a food source. The indoor temperature was 26 ± 2 °C, and the humidity was 70 ± 10%. The temperature and humidity during rearing were consistent with those during the experiment.

### 2.3. Synthesis and Characterization of Nanomaterials

#### 2.3.1. Synthesis of Raspberry-like Silica

Based on the work of Zhang et al. [34], 1.54 g of cetyl trimethyl ammonium bromide (CTAB) and 0.3 mL of triethanolamine (TEA) were dissolved in 100 mL of distilled water at 80 °C. Next, 14.6 g of tetraethyl silicate (TEOS) was quickly added to the system by dissolution via stirring for 1 h. The stirring reaction was then continued at 1200 rpm for 2 h, followed by rapid cooling in ice water and centrifuging at 15,000× *g* for 30 min. Then, the precipitation was washed 3 times with distilled water and ethanol and centrifuged for 30 min at 10,000× *g*. The precipitation was dried overnight in a vacuum at 80 °C and weighed.

#### 2.3.2. Characterization

In total, 10 mg of nanomaterial was weighed, added to 10 mL of DMF (N, N-Dimethylformamide), and ultrasonically dispersed for 30 min. Then, 10 μL of dispersion droplets was absorbed and added to the surface of a single crystal silicon wafer and allowed to dry naturally. Lastly, gold was sprayed and observed using a Field Emission Scanning Electron Microscope (SEM) S-4800, which is located at Hainan University, Haikou, China.

### 2.4. Screening of 16 Kinds of Nanomaterials

#### 2.4.1. Contact Toxicity of Nanomaterials to Red Imported Fire Ants

A total of 16 kinds of nanomaterials, including diatomite, attapulgite, hydrotalcite, zein, copper oxide, halloysite, silica, multi-wall carbon nanotubes, graphite oxide, and graphene nanosheets, were mixed with corn meal and an attractant in a ratio of 1:8:1 by weight (corn meal: attractant: nanomaterial = 8:1:1) and configured into 10% nano-bait. No nanomaterials were added to the control diet, to which corn flour was added. Next, 1 g of nanometer bait was added to the bottom of a 250 mL beaker and distributed evenly. In total, 300 worker ants were selected to crawl in a beaker for 2 h. After 2 h, 50 ants that could move normally were selected and placed into a lunch box, and 3 repetitions were set. A 10% sucrose solution was then added to feed the ants. If a worker curled up and was touched slightly with no physical reaction, it was considered dead. Larvae and reproductive ants were tested in the same way, and the number was reduced to 10. In addition, 50 worker ants were added to each lunch box to take care of the larvae and reproductive ants.

#### 2.4.2. Feeding Toxicity of Nanomaterials to Red Imported Fire Ants

Fifty small workers were randomly selected, placed in a 9 cm plastic box coated with PTFE powder (Polytetrafluoroethylene powder, to stop red imported fire ants from escaping from the inside of the box), and starved for 6 h. Sixteen types of nanomaterials were mixed with corn flour and attractants, and the weight of nanomaterials accounted for 10%. Next, we weighed 0.5 g of the prepared powder and placed it in a 3 cm Petri dish, which was then placed in the middle of a 9 cm plastic lunch box for red imported fire ants to feed. The number of workers who died was recorded every 24 h. If a worker curled up and was touched slightly with no physical reaction, it was considered dead. Corn flour was used as a control with 3 replicates per treatment. Distilled water was added to feed the ants. The same method was used for the larvae and reproductive ants, reducing the number to 10. In addition, 50 worker ants were added to each lunch box to take care of the larvae and reproductive ants.

### 2.5. Effect of High-Activity Nanomaterials on the Behavior of Red Imported Fire Ants

#### 2.5.1. Digging Behavior

Based on the work of He et al. [35], fine sand was screened with a 40-mesh sieve, dried after washing with water, dried again after 12 h, and weighed twice, with the error within 0.02 g. The toxicity of the nanomaterials was evaluated by comparing the quality of fine sand transported by red imported fire ants after being fed the high-activity nanomaterials described above for 24 h. Four 2 mL glass bottles were glued under a 15 cm plastic Petri dish for support. The distance between the glass bottles and the center of the Petri dish was 5 cm. A round hole with a diameter of 3 mm was drilled diagonally into the center of the two vials. Two glass bottles with drilled holes were filled with sand, and the other two vials without drilled holes acted as supports. The inner wall of the 15 cm Petri dish was coated with PTFE to prevent red imported fire ants from escaping. The sand used for filling was mixed with clean water in a certain proportion (7.68 mL water + 120 g sand) and used to fill a drilled vial (with 3.19 g of moist sand and about 3 g of dry sand). Fifty workers were randomly selected from 300 worker ants fed with highly active nanomaterials for 24 h and placed in Petri dishes. Distilled water was added to feed the ants. After 24 h, the sand scattered on the Petri dish was collected and dried in a drying oven until the quality remained unchanged, and the quality of the dry sand was recorded.

#### 2.5.2. Act of Abandoning Corpses

According to Ning et al. [36], the red imported fire ant workers will carry dead individuals to a pile, an act called corpse abandonment. During the test, 500 workers, 50 larvae, and 5 reproductive ants were placed in a plastic box (30 cm × 24 cm × 9 cm) with an artificial nest. After 24 h of starvation treatment, 1 g of high-activity nanomaterial bait was weighed and placed in a 3 cm Petri dish, which was itself placed 5 cm from the artificial ant nest, and each nest was fed with high-activity nanomaterials for 24 h. Distilled water was added to feed the ants. After 24 h, the food was removed, and 40 dead workers were placed 5 cm away from the nest. We recorded the time taken by workers to find the body, begin moving the body, and remove all the carcasses.

#### 2.5.3. Foraging Behavior

Based on the work of Huang et al. [37], we evaluated the ability of the colony to feed. Highly active nanomaterials were used to feed the test ants. 500 workers, 50 larvae, and 5 reproductive ants were placed in a plastic box (30 cm × 24 cm × 9 cm) with an artificial nest. After 24 h of starvation treatment, each nest was fed with high-activity nanomaterials for 24 h; the feeding method was the same as that described in Section 2.4.2. After 24 h, the high-activity nanomaterials were removed, and 1 g of ham sausages was added. The ham sausages were replaced (taking care to remove the red imported fire ants attached to the ham sausage) every 24 h to maintain the attractiveness of the food. We weighed the replaced ham sausages, calculated the weight of the ham sausages to be eaten, and placed the ham sausages in a plastic box without any ants to remove the effects of natural air drying. At the same time, we also constantly replenished the water and recorded the time it took for the first worker ant to find the food, as well as the time it took for the workers to start carrying the food. We then recorded the number of red imported fire ants on the food within 60 min of starting to add ham sausages.

### 2.6. Composite Activity of Nanomaterials and Indoxacarb

The compound baits were configured according to the weight ratios in Table 1. No nanomaterials or indoxacarb were added to the blank control baits, while another control containing indoxacarb and no nanomaterials was also set.

The other experimental steps were the same as those in Section 2.4.

### 2.7. Effects of a Combination of Nanomaterials and Indoxacarb on the Behavior of Red Imported Fire Ants

The compound baits were configured according to the weight ratios in Table 1. The content of indoxacarb was adjusted to 0.0025% and 0.001%, and the content of the nanomaterials was 10 times that of indoxacarb. No nanomaterials or indoxacarb were added to the blank control baits, while another control containing indoxacarb and no nanomaterials was also set.

The other experimental steps were the same as those in Section 2.5.

### 2.8. Statistical Analyses

All data, including activity and effects on behavior, were provided as the mean + standard error (SE) of three replicates. The normality of the data was checked using Proc Univariate (SAS 9.4, SAS Institute, Cary, NC, USA), and the results were analyzed via a one-way analysis of variance (ANOVA). Tukey’s HSD test was used to separate the means at *p* ≤ 0.05. To perform the analyses, SPSS version 26 was used (SPSS Inc., Chicago, IL, USA). Graphs were generated using Origin 2022 (OriginLab Corporation, Northampton, MA, USA).

## 3. Results

### 3.1. Activity of 16 Kinds of Nanomaterials against Red Imported Fire Ants

As show in Table 2 and Table 3, the contact and feeding toxicity of 16 nanomaterials against workers, juveniles, and reproductive red imported fire ants were tested by mixing the nanomaterials, corn flour, and attractants into baits. The results showed that after 72 h of contact treatment, diatomite, silicon dioxide, and multi-walled carbon nanotubes (short) were relatively good, and the mortality of workers reached 98.67%, 97.33%, and 68%, respectively. After 72 h of feeding treatment, it was found that multi-walled carbon nanotubes (short), raspberry-like silica, and Halloysite (200 nm) were relatively effective, and the mortality rate of workers reached 16.67%, 12.67%, and 12.67%, respectively. After 72 h, the mortality of larvae and reproductive ants was lower than 17% and 10%, respectively, after exposure and feeding treatment. Considering the toxicity of each species and the difficulty of material acquisition, diatomite, silica (raspberry-shaped), and multi-walled carbon nanotubes (short) were selected for follow-up experiments.

### 3.2. Characterization of Nanomaterials

As show in Figure 1, three selected nanomaterials, diatomite, raspberry-like silica, and multi-walled carbon nanotubes were observed via scanning electron microscopy (SEM). As shown in Figure 1, Diatomite is a loose and porous modified material, with a size of about 10 μm. Silica (raspberry-shaped) has good morphology and uniform size, ranging from 60 nm to 80 nm. The multi-walled carbon nanotubes are slender, with a thickness of about 50 nm and a length of 0.5–2 μm.

### 3.3. Effect of High-Activity Nanomaterials on the Behavior of Red Imported Fire Ants

As shown in Figure 2, after 10% diatomite was fed for 24 h, the excavation volume decreased compared with that of the control group. After 10% silica (raspberry-shaped) was fed, the amount of excavation decreased slightly, but there were no significant differences compared with the control group. After 10% multi-walled carbon nanotubes (short) were fed for 24 h, the excavation amount increased compared with that of the control group. However, there were no significant differences compared with the control group.

As shown in Figure 3, after feeding the three kinds of nanomaterials for 24 h, the time required for red imported fire ants to find the dead body, begin to move the dead body, and move the whole body increased. Diatomite significantly inhibited the behavioral indexes of red imported fire ants in disposing of the dead. Raspberry-like silica significantly extended the time required for workers to find the dead body, but the time required for workers to carry the dead body was not significantly different from that of the control. Treatment of multi-walled carbon nanotubes had no significant effect on the time required for workers to find dead bodies but significantly extended the time for red imported fire ants to carry dead bodies. Treatment with the three nanomaterials extended the time required for workers to find and transport the dead bodies. The dumping behavior of red imported fire ants was inhibited after being fed the nanomaterials. 

As shown in Figure 4, after the three kinds of nanomaterials were fed for 24 h, the time required for workers to find food and start to transport the food was significantly increased, however after treatment with multi-walled carbon nanotubes, the time required for red imported fire ants to start to transport food was not significantly different from that of the control group, which may be due to the fact that feeding with nanomaterials reduced the activity and gathering speed of red imported fire ants. In addition, after feeding the three kinds of nanomaterials, the number of foraging workers in the first 20 min of normal feeding decreased significantly compared with that of the control, as did the foraging speed. Compared with the control group, the number of foraging workers in the diatomite treatment group decreased significantly, which reduced the vitality of the ant colony. Only the total feeding amount of the workers on the first day was higher. Furthermore, the diatomite treatment significantly reduced the food intake of the workers, and the subsequent food intake had no significant inhibition effect, which was not significantly different from the results of the control.

### 3.4. Composite Activity of Nanomaterials and Indoxacarb

As shown in Figure 5, the contact activities of three kinds of nanomaterials combined with indoxacarb on red imported fire ants were tested. Compared with the positive control, the contact toxicity of nanomaterials combined with bait increased among the workers, and most of the workers died at 12 h.

For juvenile ants, the combination of nanomaterials and indoxacarb increased the contact toxicity among juvenile ants compared with that of the indoxacarb control. When 5% of the nanomaterials was combined with 0.05% of indoxacarb, almost all the young ants died after 72 h of treatment. Compared with the indoxacarb control, the toxicity of the three nanomaterials combined with indoxacarb among workers increased significantly.

In the treatment using diatomite combined with indoxacarb, there were significant differences in all concentrations between the treatment and indoxacarb control group; the death rate of juvenile ants also increased. Under treatment with raspberry-like silica, there was no significant difference between the 0.5% treatment group and the control group, and the 2.5% and 1% treatment groups significantly increased the mortality of juvenile ants after 48 h. In the multi-walled carbon-nanotube treatment group, there was no significant difference between the 0.5% treatment group and the control group, and the 2.5% and 1% treatment groups significantly increased the mortality of juvenile ants compared with that in the indoxacarb control group. Compared with the three nanomaterials, the composite effect of diatomite was relatively better, followed by that of multi-walled carbon nanotubes.

As shown in Figure 6, a combination of nanomaterials and indoxacarb increased the feeding toxicity of the combined bait among workers compared with that of the indoxacarb control, with a significant difference observed after 24 h. With an increase in nanomaterial content, feeding toxicity increased but with no significant difference. After 48 h of treatment, the three kinds of nanomaterials killed most of the workers.

Using a combination of nanomaterial and indoxacarb increased the feeding toxicity among juvenile ants compared with that of the indoxacarb control. Compared with the indoxacarb control, the toxicity of 5% nanomaterials and 0.05% indoxacarb among workers increased significantly. After four days of combined treatment with diatomite and indoxacarb, the infant mortality of each treatment group increased significantly compared with that of the control group. Raspberry-like silica significantly increased infant mortality at 5% and 2.5% concentrations, but there was no significant difference in infant mortality at lower concentrations compared with that of the indoxacarb control group. The infant mortality rate in the multi-walled carbon-nanotube treatment group was significantly higher than that in the indoxacarb control group after 2 d, but there was no significant difference between the low concentration (0.5%, 1%) and indoxacarb control group at 4 d and 5 d. Compared with the three nanomaterials, the composite effect of diatomite and multi-wall carbon nanotubes was found to be relatively good.

### 3.5. Effects of Combination of Nanomaterials and Indoxacarb on the Behavior of Red Imported Fire Ants

As shown in Figure 7, after being fed the supplemental diet for 24 h, the excavation amount decreased significantly compared with that of the control group. Compared with the indoxacarb control group, the three kinds of nanomaterial treatment groups also significantly reduced the number of digging workers. When the concentration of indoxacarb was 0.0025%, diatomite offered a better inhibition effect than that of the other two nanomaterials.

As shown in Figure 7, after being fed the three kinds of nanomaterials for 24 h, the corpse-removal behavior of red imported fire ants was inhibited. Treatment with the three nanomaterials extended the time required for workers to find and transport the dead bodies. There was no significant difference in the time of body discovery between the three material treatments and the indoxacarb control. The combination of the three nanomaterials and indoxacarb significantly extended the time required for workers to start carrying the dead. The three materials all extended the total time required for workers to abandon the body, but only diatomite and multi-walled carbon nanotubes were significantly different from indoxacarb, with relatively good effects.

As shown in Figure 8, after being fed the three kinds of nanomaterials for 24 h, the time required for workers to find food and start to transport food increased. Additionally, the time required for food discovery and transport was more significantly increased in the diatomite and multi-wall carbon-nanotube treatment group compared with that in the indoxacarb control group, with no significant difference between the raspberry-like silicon dioxide and pharmaceutical control group. Feeding with nanomaterials may have reduced the activity and aggregation rate of red imported fire ants. In addition, the number of foraging workers in the diatomite-treated group was significantly lower than that in the blank control group within 1 h after feeding. In the raspberry-like silica group, the number of workers foraging in the first 40 min was significantly higher than that in the other two material treatment groups, with no significant difference between the two groups and the blank control group. The number of foraging workers in the multi-walled carbon nanotube treatment group was significantly lower than that in the blank control group in the first 40 min, but there was no significant difference between this group and the blank control after 40 min. The food intake of workers in the three nanomaterial treatment groups decreased compared with that observed in the indoxacarb control but without a significant difference. Moreover, compared with the blank control, food intake was significantly reduced. After treatment with the three materials, the foraging speed of the ant colony decreased, the number of foraging workers decreased, and the vitality of the ant colony was reduced.

## 4. Discussion

Nanomaterials offer numerous advantages in the field of pest management that exceed those of traditional pesticide products. In recent years, there has been a surge of research interest in developing nanomaterial-based pesticides to enhance the biological efficacy of pesticides. Several studies have demonstrated the potent biological activity of cationic nano-chitin whiskers against wheat aphids. In addition, when combined with chemical pesticides, the dosage of both agents can be reduced by half without a significant decrease in activity. The presence of nano-chitin whiskers greatly enhances the effectiveness of chemical insecticides in killing insects while also reducing the rates of pupation and emergence [38]. The MON@CeO_2_ nanohybrid, based on CeO_2_, functions as an inhibitor of reactive oxygen species (ROS) that greatly enhances the toxicity of nitenpyram, sulfadiazine, and clothianidin, which are brown planthopper-resistant insecticides in both laboratory and field settings. Additionally, this nanohybrid effectively increases the sensitivity of pests to insecticides [39]. A mixture of silver nanoparticles and Thiram has a synergistic effect that leads to a loss of phase contrast intensity and rupture of the lipid bilayers [40]. Graphene oxide has an excellent synergistic effect on β-cyfluthrin, Monosultap, and Imidacloprid, which improves the contact toxicity against lepidoptera pests and may cause physical damage to the insect body surface, resulting in rapid water loss. In addition, the insect’s damaged body surface provides a new channel for insecticide penetration [41]. Tungsten oxide (WO_3_), iron oxide (magnetic nanoparticles, MNP), and copper-doped iron oxide (MNP-Cu) nanocomposites did not show any toxicity against the different stages of *S. littoralis*, and the pupation rate and emergence rate were significantly reduced after mixing with Cyromazine [42]. 

The results of this study demonstrated that both diatomite and multi-walled carbon nanotubes have particular insecticidal effects on red imported fire ant workers. Following 72 h of treatment, the mortality rates among the workers were recorded as 98.67% and 68%, respectively. The silica material also showed certain insecticidal activities. The contact toxicity of 20 nm and 410 nm silicon dioxide among workers also reached 97%. Nevertheless, the toxicity of nanomaterials when consumed by workers was minimal, with each case measuring below 17% after a period of 72 h. Furthermore, the tested nanomaterials had a minimal impact on both juvenile and reproductive ants, with mortality rates consistently below 17%. Thus, solely relying on nanomaterials to manage red imported fire ants proves challenging. Given the challenges associated with acquiring nanomaterials and the expenses involved in their regulation, diatomite, Silica (in a raspberry-shaped form), and multi-walled carbon nanotubes were chosen and combined with indoxacarb. This study revealed that the efficacy of combining diatomite with indoxacarb in controlling red imported fire ants was higher than that in the indoxacarb control group. After 12 h of contact treatment, most of the workers died. After 72 h of exposure treatment, the mortality of larvae was significantly higher than that of the indoxacarb control group. The feeding toxicity to workers and larvae also increased significantly, which was significantly different from that observed in the indoxacarb control. Scanning electron microscopy analysis revealed that the diatomite edge had a jagged morphology and was able to destroy the body walls of red imported fire ants. The destruction of the body walls of worker ants may have been a cause of death among red imported fire ants. Furthermore, disruption of the body wall facilitated the penetration of indoxacarb, thereby enhancing the efficacy of indoxacarb in controlling the target.

Red imported fire ants are social insects with complex group behavior and a clear division of labor among groups [43]. Ant colonies have different behaviors, which jointly maintain the development of ant colonies. Through the observation of ant colony behavior, the overall situation of an ant colony can be determined. As a kind of mound-building species, digging soil is an inherent behavior of red imported fire ants, which can be used to determine the expansion or migration capabilities of ant nests and the construction of tunnels [44]. Red imported fire ants dispose of their bodies to prevent the spread of pathogens and reduce the risk of disease in their moist, enclosed nests [45]. The removal of dead bodies from the nest, also known as necro migration, is one of the best-described behaviors of social insects [46]. The foraging behavior of red imported fire ants is a complex behavior in which tasks are assigned to numerous individuals [47,48]. 

The effect of treatment on the foraging behavior of ants can be reflected by statistics on the quantity and food intake of foraging ants. Through an evaluation of ant colony behavior, the overall vitality level of an ant colony can be determined, and the influence of applying chemicals to the ant colony can be analyzed from multiple perspectives to better reflect the control effects.

The results of this study showed that feeding red imported fire ants three kinds of nanomaterials could inhibit the behavior of the ants. After diatomite treatment, the number of digging workers decreased. Treatment with the three nanomaterials inhibited the discarding and foraging behaviors of the workers, significantly increased the time required for these behaviors, and significantly decreased the number of foraging workers in the first 20 min. The combination of indoxacarb, diatomite, and multi-walled carbon nanotubes significantly inhibited various behaviors of the ant colonies and improved the effects of indoxacarb.

After being fed food containing nanomaterials combined with indoxacarb, the digging behavior of red imported fire ants was inhibited, which weakened the expansion and migration abilities of the ant colony and inhibited the spread of red imported fire ants to a certain extent. Inhibiting the dumping behavior among red imported fire ants and prolonging the time required to remove dead workers would make bacteria more likely to breed in the ant nest and require that the ant colony spend more energy to combat this issue. In addition, the agent can be transmitted to normal workers through contact with cadavers [49], and the extension of the corpse-removal time makes it possible for the agent to be transmitted to more workers, thereby exerting a more thorough control effect. The prolongation of foraging time and a decrease in the number of foraging workers inhibited the foraging behavior of the ant colony, which negatively impacted normal reproduction and diffusion activities and thus inhibited the overall activities of the ant colony. In addition, a decrease in various behavioral abilities reflects the overall vitality of the ant colony after feeding, which reduces the competitiveness of the ant colony under natural conditions and inhibits the spread of red imported fire ants.

According to existing research results, diatomite and multi-walled carbon nanotubes can enhance the activity of indoxacarb against red imported fire ants and inhibit the behavior of ant colonies, possibility due to the nanomaterials destroying the body walls of workers and increasing the entry route of indoxacarb. Diatomite and multi-walled carbon nanotubes can be added as additives to the control agents to improve the control effects of red imported fire ants, reduce the use of indoxacarb, and delay the development of resistance.

In this study, only a limited number of ants were subjected to experiments in the laboratory. These conclusions need to be further verified by field experiments.

## 5. Conclusions

The present findings demonstrated that diatomite and multi-walled carbon nanotubes exerted a specific regulatory influence on red imported fire ant workers, effectively suppressing certain behaviors within the ant colony. Furthermore, when indoxacarb was used in conjunction with indoxacarb, it notably intensified the effectiveness of indoxacarb against red imported fire ants and severely impeded the behavior of the ant colony. The incorporation of diatomite and multi-walled carbon nanotubes in the control agent enhanced the efficacy of controlling red imported fire ants, decreased reliance on indoxacarb, and prolonged the onset of resistance. This study provides evidence for the utilization of nanomaterials in the domain of red imported fire ant management.

## Figures and Tables

**Figure 1 insects-15-00096-f001:**
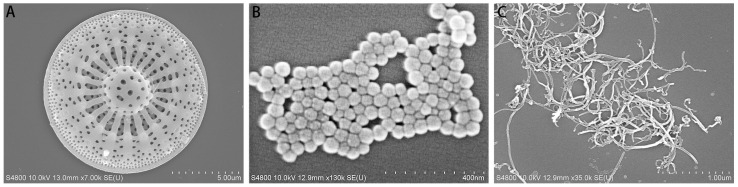
Scanning electron microscope images of three nanomaterials ((**A**): Diatomite; (**B**): Silica (raspberry-shaped); (**C**): Multi-walled carbon nanotubes).

**Figure 2 insects-15-00096-f002:**
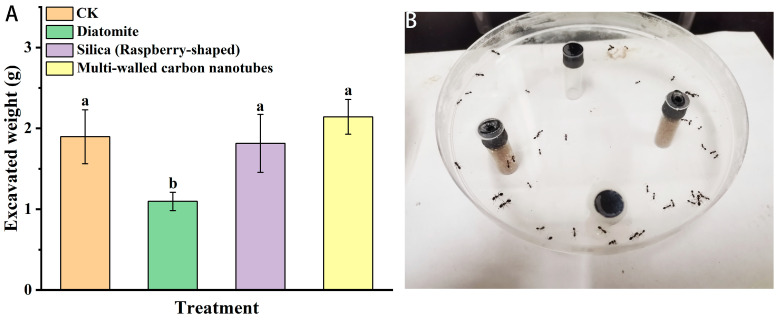
Effect of nanomaterials on the digging behavior of red imported fire ants: (**A**) Bioassay arenas. (**B**) Excavated weight of the three nanomaterials after 24 h of treatment. Data are presented as mean ± standard error (S.E.). Different letters above bars indicate significant differences in behavior-altering among treatments due to nanomaterials effects at *p* < 0.05 level based on Tukey’s honestly significant difference (HSD) test (*n* = 3).

**Figure 3 insects-15-00096-f003:**
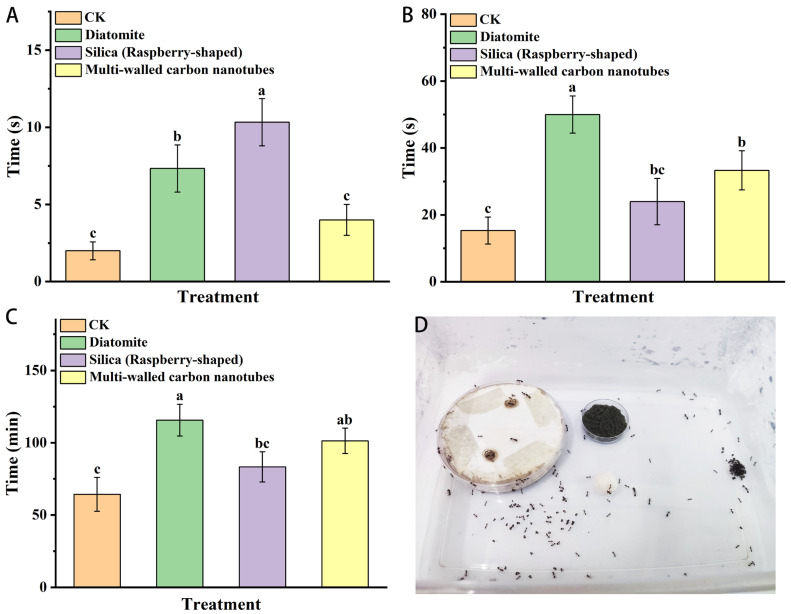
The effect of nanomaterials on the dead-dumping behavior of red imported fire ants: (**A**) The time required for workers to find dead bodies. (**B**) The time to start carrying the body. (**C**) The time to move all the dead bodies. (**D**) Test tracking site for dead-dumping behavior. Data are presented as mean ± standard error (S.E.). Different letters above bars indicate significant differences in behavior-altering among treatments due to nanomaterials effects at *p* < 0.05 level based on Tukey’s honestly significant difference (HSD) test (*n* = 3).

**Figure 4 insects-15-00096-f004:**
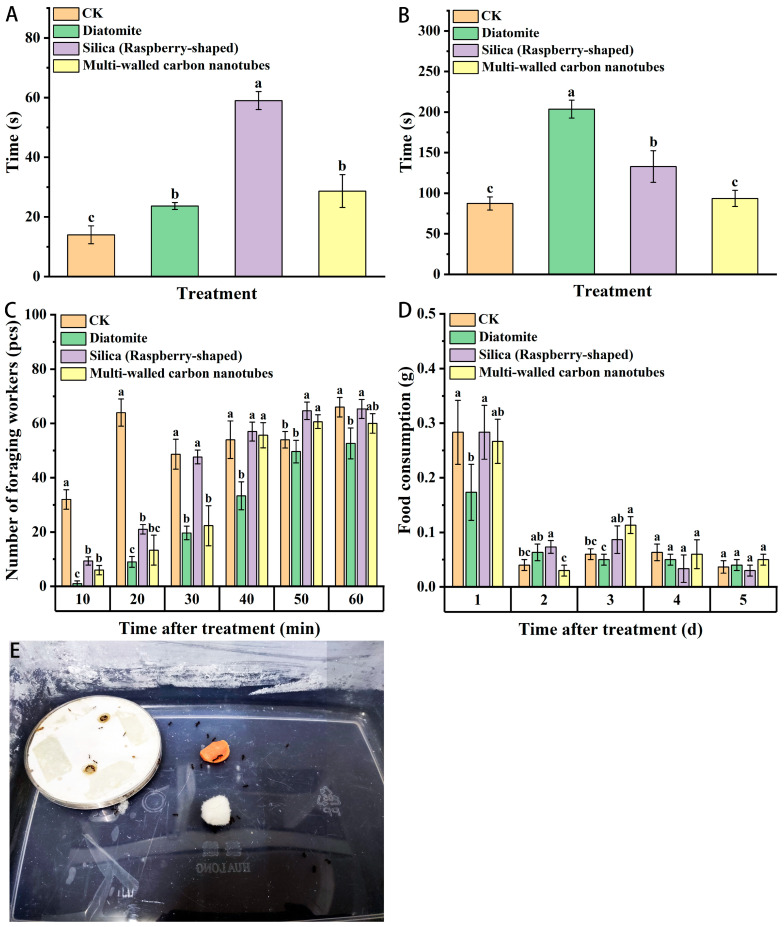
Effects of nanomaterials on the foraging behavior of red imported fire ants: (**A**) The time required for workers to find food. (**B**) The time to start carrying food. (**C**) The number of workers carrying food. (**D**) Weight of food carried by workers. (**E**,**F**) Photographs of the red imported fire ant foraging behavior test site. Data are presented as mean ± standard error (S.E.). Different letters above bars indicate significant differences in behavior-altering among treatments due to nanomaterials effects at *p* < 0.05 level based on Tukey’s honestly significant difference (HSD) test (*n* = 3).

**Figure 5 insects-15-00096-f005:**
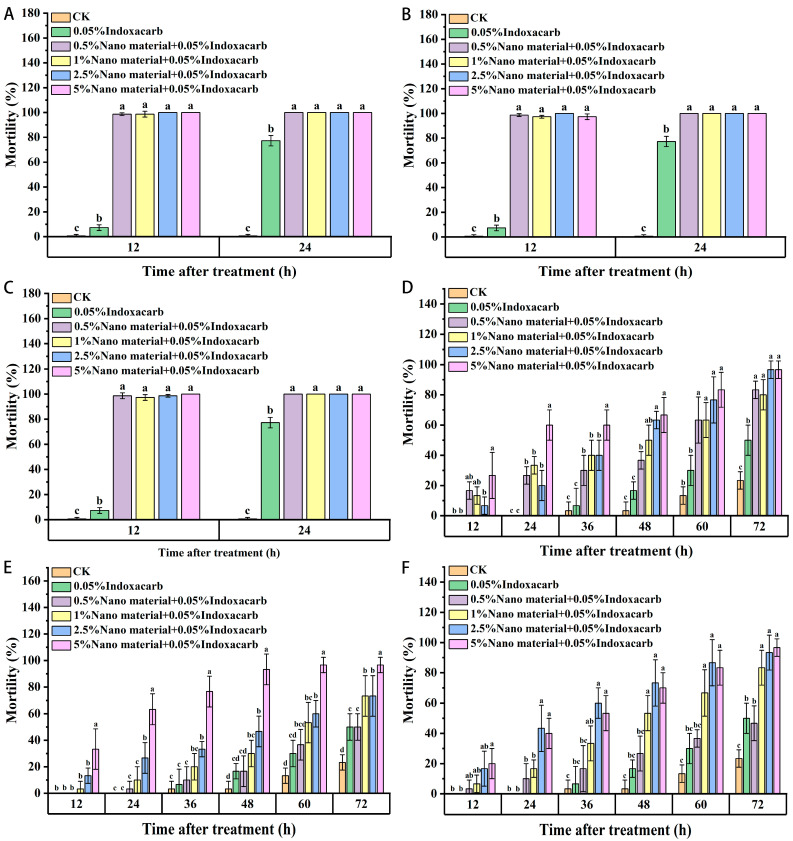
Contact toxicity of red imported fire ant workers and larvae in combination with indoxacarb and nanomaterials ((**A**–**C**): workers. (**A**): Diatomite; (**B**): Silica (raspberry-shaped); (**C**): Multi-walled carbon nanotubes. (**D**–**F**): larvae. (**D**): Diatomite; (**E**): Silica (raspberry-shaped); (**F**): Multi-walled carbon nanotubes). Data are presented as mean ± standard error (S.E.). Different letters at each recording time point indicate significant differences among treatments at *p* < 0.05 level based on Tukey’s HSD test (*n* = 3).

**Figure 6 insects-15-00096-f006:**
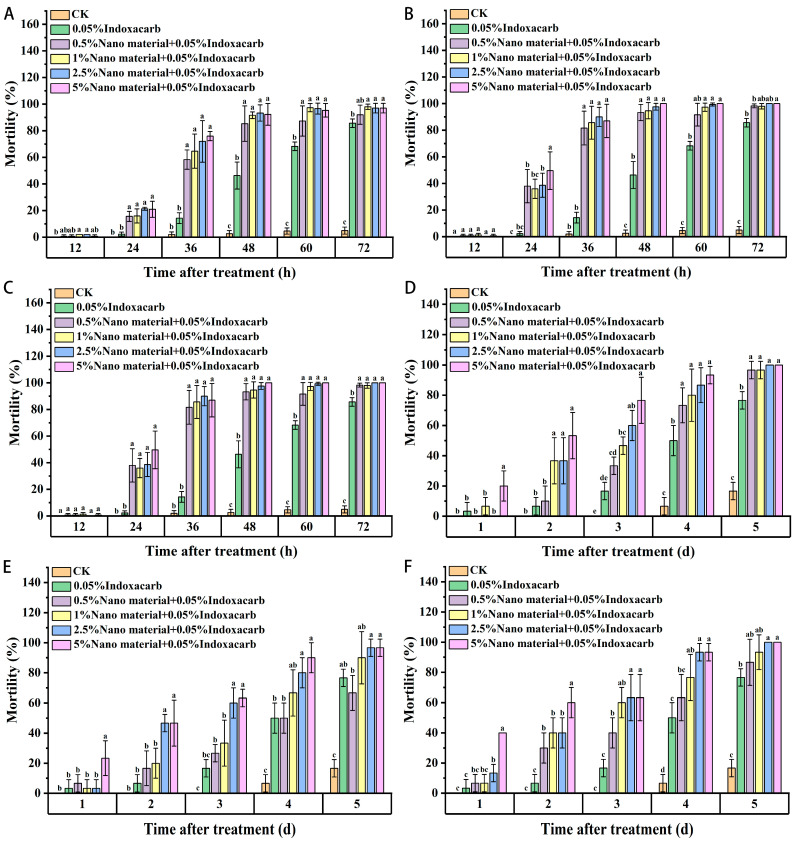
Feeding toxicity of red imported fire ant workers and larvae in combination with indoxacarb and nanomaterials ((**A**–**C**): workers. (**A**): Diatomite; (**B**): Silica (raspberry-shaped); (**C**): Multi-walled carbon nanotubes. (**D**–**F**): larvae. (**D**): Diatomite; (**E**): Silica (raspberry-shaped); (**F**): Multi-walled carbon nanotubes). Data are presented as mean ± standard error (S.E.). Different letters at each recording time point indicate significant differences among treatments at *p* < 0.05 level based on Tukey’s HSD test (*n* = 3).

**Figure 7 insects-15-00096-f007:**
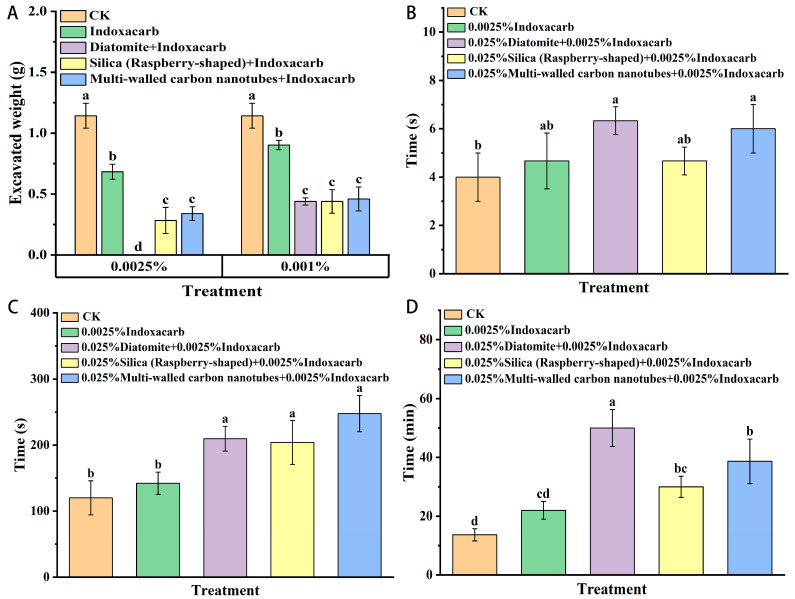
Effect of the combination of indoxacarb and nanomaterials on the digging and corpse-removal behavior of red imported fire ants ((**A**): the weight of the sand excavated by the workers; (**B**): the time required for workers to find dead bodies; (**C**): The time to start carrying the body; (**D**): The time to move all the dead bodies). Data are presented as mean ± standard error (S.E.). Different letters above bars indicate significant differences in behavior-altering among treatments due to nanomaterials effects at *p* < 0.05 level based on Tukey’s honestly significant difference (HSD) test (*n* = 3).

**Figure 8 insects-15-00096-f008:**
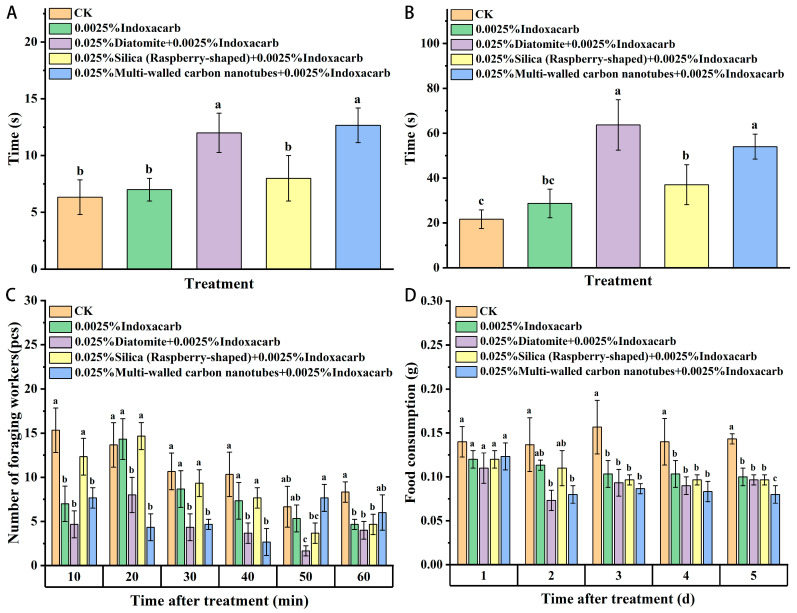
Effects of nanomaterials on the foraging behavior of red imported fire ants ((**A**): The time required for workers to find food; (**B**): The time to start carrying food; (**C**): The number of workers carrying food; (**D**): Weight of food carried by workers). Data are presented as mean ± standard error (S.E.). Different letters above bars indicate significant differences in behavior-altering among treatments due to nanomaterials effects at *p* < 0.05 level based on Tukey’s honestly significant difference (HSD) test (*n* = 3).

**Table 1 insects-15-00096-t001:** The compound formulation of nanomaterial and indoxacarb.

Component	Content
Corn flour	make up to 100%
White granulated sugar	10%
Attractants (chicken bone meal: fish meal = 1:1)	10%
Peanut oil	10%
Indoxacarb	0.05%
Nanomaterial	0.5%, 1%, 2.5%, 5%

**Table 2 insects-15-00096-t002:** Mortality of red imported fire ant workers after contact treatment with 16 kinds of nanomaterials.

Nanomaterials	Mortality (%)		
	24 h	48 h	72 h
CK	0.67 ± 0.67 d	2.00 ± 1.15 g	4.00 ± 1.15 d
Diatomite	92.67 ± 7.33 a	94.67 ± 5.33 a	98.67 ± 1.33 a
Attapulgite	39.3 ± 10.35 bc	42.67 ± 10.73 bc	46.67 ± 13.78 bc
Zein	3.33 ± 0.67 d	3.33 ± 0.67 fg	3.33 ± 0.67 d
Hydrotalcite	6.00 ± 1.15 d	17.33 ± 0.67 defg	48.67 ± 17.29 bc
Copper oxide (40 nm)	21.33 ± 7.06 cd	35.33 ± 1.76 cd	41.33 ± 3.71 bc
Copper oxide (100–200 nm)	20.67 ± 7.69 cd	24.67 ± 8.67 cdefg	28.00 ± 11.02 cd
Halloysite (200 nm)	7.33 ± 1.33 d	10.67 ± 0.67 efg	12.00 ± 1.15 d
Halloysite (500 nm)	6.00 ± 3.06 d	8.00 ± 3.46 fg	8.00 ± 3.46 d
Silica (20 nm)	92.00 ± 8.00 a	96.67 ± 3.33 a	97.33 ± 2.67 a
Silica (80 nm)	23.33 ± 2.40 cd	33.33 ± 6.57 ce	48.00 ± 9.45 bc
Silica (410 nm)	92.67 ± 7.33 a	93.33 ± 6.67 a	96.67 ± 3.33 a
Silica (raspberry-shaped)	10.00 ± 2.00 d	10.67 ± 1.76 efg	11.33 ± 1.76 d
Multi-walled carbon nanotubes (long)	22.67 ± 5.33 cd	25.33 ± 3.71 cdef	28.00 ± 3.06 cd
Multi-walled carbon nanotubes (short)	49.33 ± 19.06 b	63.33 ± 18.56 b	68.00 ± 17.01 b
Graphene nanosheets	0.67 ± 0.67 d	3.33 ± 0.67 fg	4.00 ± 1.15 d
Graphite oxide	1.33 ± 0.67 d	3.33 ± 2.40 fg	3.33 ± 2.40 d

Data are presented as mean ± standard error (S.E.). Different letters at each recording time point indicate significant differences among treatments at *p* < 0.05 level based on Tukey’s HSD test (*n* = 3).

**Table 3 insects-15-00096-t003:** Mortality of red imported fire ant workers after being fed 16 kinds of nanomaterials.

Nanomaterials	Mortality (%)		
	24 h	48 h	72 h
CK	1.33 ± 0.67 de	3.33 ± 1.76 cd	6.00 ± 1.15 bc
Diatomite	5.33 ± 1.33 abcde	7.33 ± 0.67 abcd	9.33 ± 0.67 bc
Attapulgite	6.67 ± 1.76 abc	8.00 ± 1.15 abcd	8.67 ± 0.67 bc
Zein	0.67 ± 0.67 e	2.67 ± 1.76 d	4.00 ± 2.31 c
Hydrotalcite	2.67 ± 0.67 cde	4.67 ± 1.33 bcd	8.00 ± 2.00 bc
Copper oxide (40 nm)	4.67 ± 0.67 abcde	5.33 ± 0.67 bcd	7.33 ± 1.33 bc
Copper oxide (100–200 nm)	8.00 ± 1.15 ab	10.00 ± 1.15 ab	12.00 ± 1.15 ab
Halloysite (200 nm)	3.33 ± 0.67 bcde	6.00 ± 2.00 bcd	12.67 ± 2.40 ab
Halloysite (500 nm)	8.00 ± 2.31 ab	9.33 ± 2.40 abc	11.33 ± 1.33 ab
Silica (20 nm)	5.33 ± 2.40 abcde	6.67 ± 2.91 abcd	10.67 ± 2.91 abc
Silica (80 nm)	6.00 ± 3.06 abcd	7.33± 2.40 abcd	9.33 ± 2.40 bc
Silica (410 nm)	4.67 ± 1.33 abcde	4.67 ± 1.33 bcd	8.67 ± 1.76 bc
Silica (raspberry-shaped)	5.33 ± 1.33 abcde	7.33 ± 1.76 abcd	12.67 ± 1.76 ab
Multi-walled carbon nanotubes (long)	6.67 ± 0.67 abc	10.67 ± 0.67 ab	12.00 ± 1.15 ab
Multi-walled carbon nanotubes (short)	9.33 ± 0.67 a	12.67 ± 1.33 a	16.67 ± 4.06 a
Graphene nanosheets	4.00 ± 0.00 bcde	4.67 ± 0.67 bcd	6.00 ± 1.15 bc
Graphite oxide	2.00 ± 2.00 cde	3.33 ± 3.33 cd	6.00 ± 2.00 bc

Data are presented as mean ± standard error (S.E.). Different letters at each recording time point indicate significant differences among treatments at *p* < 0.05 level based on Tukey’s HSD test (*n* = 3).

## Data Availability

All data generated or analyzed during this study are included in this published article.

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
