# Peer review of "Toxicity and Behavior-Altering Effects of Three Nanomaterials on Red Imported Fire Ants and Their Effectiveness in Combination with Indoxacarb"

_insects, 2024, doi:10.3390/insects15020096_

Round 1
Reviewer 1 Report
Comments and Suggestions for Authors
Manuscript reports the results of a laboratory evaluations of using various nanoparticles by themselves and in combination with indoxacarb against Solenopsis invicta.
Solenopsis invicta has an official Entomological Society of America common name of red imported fire ant. This common name is consistently used in journal articles internationally. Authors should use this common name in this manuscript.
The development of pesticide resistance was discussed with the references cited of #23 & #24 not discussing pesticide resistance, nor insecticide resistance in ants.
Statistical analyses should be described more fully, including the use of transformations if needed to meet assumptions of analysis of variance. In addition, statistical tests should be included in Tables and Figures.
Tables 2 and 3: add footnote describing statistical test used. Are these means being presented? Are comparisons within a column?
It is important to consider the duration of the change in behavior of the colony. The behavioral change was measured for 72 hours in this study. Perhaps the colony could recover. The behaviors documented in this study could reflect mortality of the treatments to workers. Was any data recorded on worker mortality or the number of live ants remaining after the duration of the tests, or past 72 hours? This would provide a more direct measure of the effects of the treatments on the colony or colony fragments used in this study.
Indicate how it was confirmed that ants fed on the nanomaterials bait as opposed to being exposed by contacting the bait on the ants' tarsi.
The manuscript describes laboratory studies with limited numbers of ants. Authors should qualify their conclusions that these results need further study with large colonies and field evaluations.
Comments on the Quality of English LanguageDiction and grammar throughout the manuscript must be improved. Currently, there are too many grammatical errors in the manuscript which made several sentences awkward to read and/or difficult to interpret. Some of these texts were highlighted in yellow without comments. Note that not all grammatical or diction errors were indicated in manuscript.
Reviewer 2 Report
Comments and Suggestions for Authors
This manuscript presents an intriguing study on the synergistic effects of nanoparticles combined with indoxacarb for controlling red fire ants, a topic of significant importance for pest management. I commend the authors on their work and have a few suggestions that could enhance the clarity and quality of the manuscript:
1.The resolution and clarity of all eight figures presented are insufficient for a thorough evaluation. High-quality figures are essential for publication and understanding the results; therefore, I recommend that the authors provide images with improved resolution.
2. The current description of the experimental setup in the Materials and Methods section is not detailed enough to replicate the study, which is a cornerstone of scientific research. Specifically, detailed temperature and humidity conditions during testing should be included, as they are known to affect the efficacy of pest control methods. For instance, Chen et al. (2021) observed that increased humidity affects insect mortality rates. Were the tests conducted in the insectary? This information is vital for interpreting the results (https://doi.org/10.1111/mve.12521).
Line 146: Please define DMF.
Line 154: Clarify whether the '8:1:1' ratio of corn meal, attractant, and nanomaterial is by weight or volume. Please also clarify all the ratios throughout the manuscript.
Line 159: Specify the standard diet used to "feed them normally."
Line 164: Define PTFE within the context of its use in the study.
Line 167: Do you mean the number of dead ants was recorded?
Lines 168-169: "if the workers curl up and touches the worker ant slightly" to " if a work curls up and is touched slightly"
Line 198: "fed" to "feed"
Line 203: Explain the methodology used to assess the quality of the consumed ham.
Line 207: Describe the procedure for removing ants from the ham during weighing to ensure accuracy.
3. The protocols described in sections 2.4.1 and 2.4.2 raise concerns about the validity of the contact toxicity assay and feeding toxicity assay. It seems that the ants were simultaneously in contact with and consuming the test materials, which could confound the results. Clarification on this aspect is necessary to understand the distinct effects of contact and ingestion.
Comments on the Quality of English Language1. Please check the formatting throughout the paper. It should be written with a space between the number and the unit, for example, it should read as "10 mg" not "10mg" in Line 146.
2. Unnecessary space between "red" and "fire" in Lines 53, 56, and 59.
Round 2
Reviewer 2 Report
Comments and Suggestions for Authors
The authors addressed my comments well. However, the resolution of all the figures is still of low quality. Some figure legends are not readable.
